# A uniform data processing pipeline enables harmonized nanoparticle protein corona analysis across proteomics core facilities

Hassan Gharibi [1,10], Ali Akbar Ashkarran[2,10], Maryam Jafari[3], Elizabeth Voke[4], Markita P. Landry [4,5,6,7], Amir Ata Saei [8,9] ✉ & Morteza Mahmoudi [2] ✉

Protein corona, a layer of biomolecules primarily comprising proteins, forms dynamically on nanoparticles in biological fluids and is crucial for predicting nanomedicine safety and efficacy. The protein composition of the corona layer is typically analyzed using liquid chromatography-mass spectrometry (LC-MS/MS). Our recent study, involving identical samples analyzed by 17 proteomics facilities, highlighted significant data variability, with only 1.8% of proteins consistently identified across these centers. Here, we implement an aggregated database search unifying parameters such as variable modifications, enzyme specificity, number of allowed missed cleavages and a stringent 1% false discovery rate at the protein and peptide levels. Such uniform search dramatically harmonizes the proteomics data, increasing the reproducibility and the percentage of consistency-identified unique proteins across distinct cores. Specifically, out of the 717 quantified proteins, 253 (35.3%) are shared among the top 5 facilities (and 16.2% among top 11 facilities). Furthermore, we note that reduction and alkylation are important steps in protein corona sample processing and as expected, omitting these steps reduces the number of total quantified peptides by around 20%. These findings underscore the need for standardized procedures in protein corona analysis, which is vital for advancing clinical applications of nanoscale biotechnologies.

Once exposed to biological fluids, the surface of nanoparticles is rapidly covered with a layer composed of ions and various types of biomolecules called biomolecular/protein corona[1]. The composition of the protein corona, in terms of the type, abundance, and decoration of participating proteins, determines how biosystems (e.g., cells) perceive nanoparticles and respond to them[2]. A recent meta-analysis of 2,134 published manuscripts in the field of protein corona revealed a great heterogeneity in the proteomics analysis of nanoparticle corona (e.g., in terms of the numbers of identified unique proteins)[3]; therefore, there is a great need for the development of standardized approaches/protocols to improve the proteomics characterization of nanoparticle protein corona across various labs/cores[4].

Mass spectrometry-based proteomics generally produces reproducible data[5], and the main difference between similar experiments

[1]Division of Chemistry I, Department of Medical Biochemistry and Biophysics, Karolinska Institutet, Stockholm, Sweden. [2]Department of Radiology and Precision Health Program, Michigan State University, East Lansing, MI, USA. [3]Division of ENT Diseases, Department of Clinical Science, Intervention and Technology, Karolinska Institutet, Stockholm, Sweden. [4]Department of Chemical and Biomolecular Engineering, University of California, Berkeley, Berkeley, CA, USA. [5]Innovative Genomics Institute, Berkeley, CA, USA. [6]California Institute for Quantitative Biosciences, University of California, Berkeley, Berkeley, CA, USA. [7]Chan Zuckerberg Biohub, San Francisco, CA, USA. [8]Centre for Translational Microbiome Research, Department of Microbiology, Tumor and Cell Biology, Karolinska Institutet, Stockholm 17165, Sweden. [9]Biozentrum, University of Basel, 4056 Basel, Switzerland. [10]These authors contributed equally: Hassan Gharibi, Ali Akbar Ashkarran. ✉e-mail: amir.saei@unibas.ch; mahmou22@msu.edu

performed in different labs would be noted in the number of proteins quantified in a given sample (i.e., proteome coverage). Therefore, in cases with low proteome coverage, the lack of detection of a low-abundant genuine target can impose a bias, and as such a less important target candidate might be selected for follow-up analysis. This issue is exemplified by the proteomics analysis of biological fluids (e.g., plasma/serum), and by extension in the analysis of (plasma/serum) protein corona, as the presence or absence of a protein in the corona layer can lead to data misinterpretation or missing a biomarker. The analysis of protein corona suffers from similar challenges as in plasma proteomics, including mainly the broad dynamic range[6], i.e., 22 proteins comprise 99% of plasma proteins by weight[7], with peptides from such proteins crowding the mass spectra and preventing the in-depth analysis of the proteome, especially for proteins with rather low abundance[8]. Furthermore, another analytical difficulty is the presence of different protein isoforms in plasma. In fact, emerging technologies using nanoparticle protein corona have been developed for reducing the biological complexity of a given biological fluid in biomarker discovery[9–11]. Despite these challenges, scientists have quantified thousands of proteins in plasma, leading to the discovery of distinct disease-based biomarkers[12–16].

While plasma proteomics is continuously improving, there are limited attempts at standardization across different proteomics studies with regard to sample preparation, as well as data extraction, cleaning, and processing[17]. However, considerations and recommendations about study design, plasma sample collection, quality assurance, sample preparation, MS data acquisition, data processing, and bioinformatics together with minimum reporting requirements for proteomics experiments have been discussed[17,18].

The quality and proteome coverage of protein corona reported by a given core facility can be affected by the sample preparation protocols, analytical columns, liquid chromatography (LC) systems, and mass spectrometry (MS) instruments, as well as the method parameters and duration of the analysis. Other sources of variation involve the platform for database search of the raw files, search settings, control of false-discovery rate (FDR), the inclusion of post-translational modifications, and the sequence database used.

To investigate the extent and source of heterogeneity in protein corona data obtained from various cores, we sent identical protein corona samples to distinct liquid chromatography–mass spectrometry (LC-MS/MS) core facilities and analyzed their reported results[4]. More specifically, 17 identical aliquots of a protein corona sample were analyzed by centers at Harvard University, Stanford University, Massachusetts Institute of Technology (MIT), Case Western Reserve University, Wayne University, University of Illinois, Cornell University, University of Tennessee, University of Nebraska-Lincoln (UNL), University of Missouri, University of Cincinnati, University of Florida, University of Kansas Medical Core (KUMC), University of Texas at San Antonio (UTSA), Michigan State University (MSU), University of California San Diego (UCSD) and University of Nevada, Reno (UNR). The analysis was performed in 3 technical replicates. No standard operating procedures were provided or requested; rather the core facilities were asked to analyze the samples according to usual practices. Hereafter, we blind the core facility names with random numbers–the same numbers as in the previous study[4], to prevent any potential conflict of interest. Essential details from the protocols can be found in Supplementary Table 1. We had requested detailed protocols from all centers, which can be found in the supplementary information of our original study[4].

In this study, we explore the influence of database search, data extraction, processing, and analysis on observed data heterogeneity. Specifically, we investigate whether employing an aggregated data-base search with uniform parameters, including controlled FDR, can help standardize and harmonize the results.

## Results

We performed a uniform database search on the LC-MS/MS raw files from 15 centers that used orbitrap detectors (the overall workflow is shown in Fig. 1). Centers 6 was excluded from the search, as the samples were analyzed by a Bruker timsTOF-PRO instrument. Center 8 was also excluded as acrylamide was used for the alkylation of cysteine (Cys) residues, while the other centers used iodoacetamide (IAA) or skipped the alkylation step. Since several centers had not specified if the reduction and alkylation of proteins had been performed, we

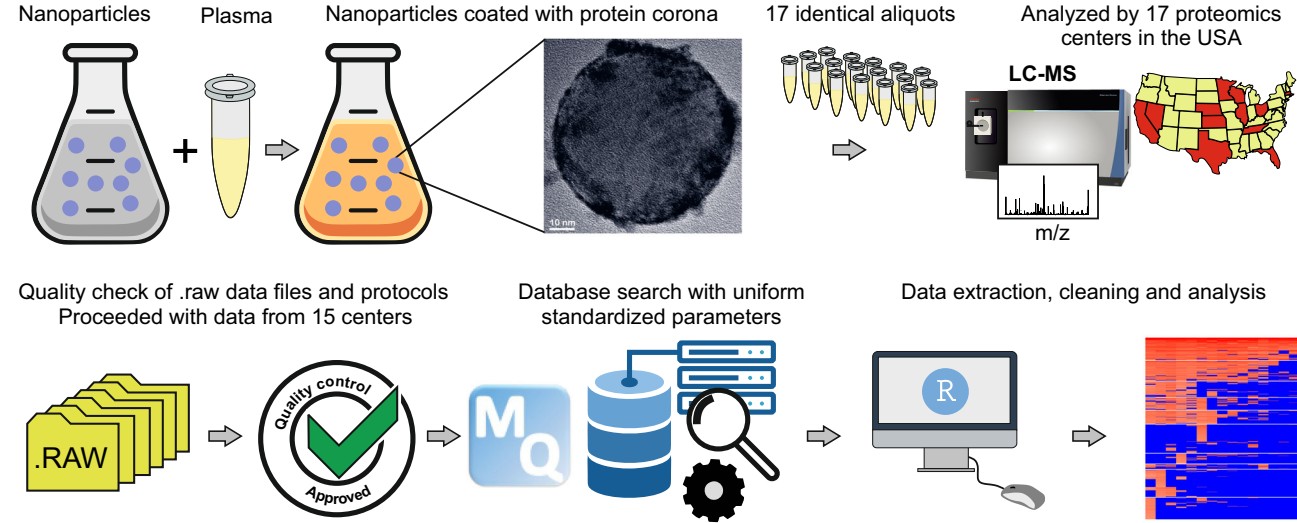

**Fig. 1 | A uniform database search on nanoparticle protein corona raw data provided by 15 core facilities.** Nanoparticles were incubated with plasma to form the protein corona. The samples were then divided into 17 identical aliquots and submitted to 17 different core facilities across the USA, as described in ref. 4. In this study, the raw files were individually examined. The protocols were also carefully examined to identify the raw files that could be processed in a uniform database search. Raw data from the 15 centers were then subjected to database search using an up-to-date version of MaxQuant with an up-to-date fasta file. Data was then extracted, cleaned, and analyzed, as described in detail below. The TEM image of the corona-coated nanoparticle is reproduced here with permission from ref. 4. *m/z* mass-to-charge ratio, LC-MS liquid chromatography coupled to mass spectrometry.

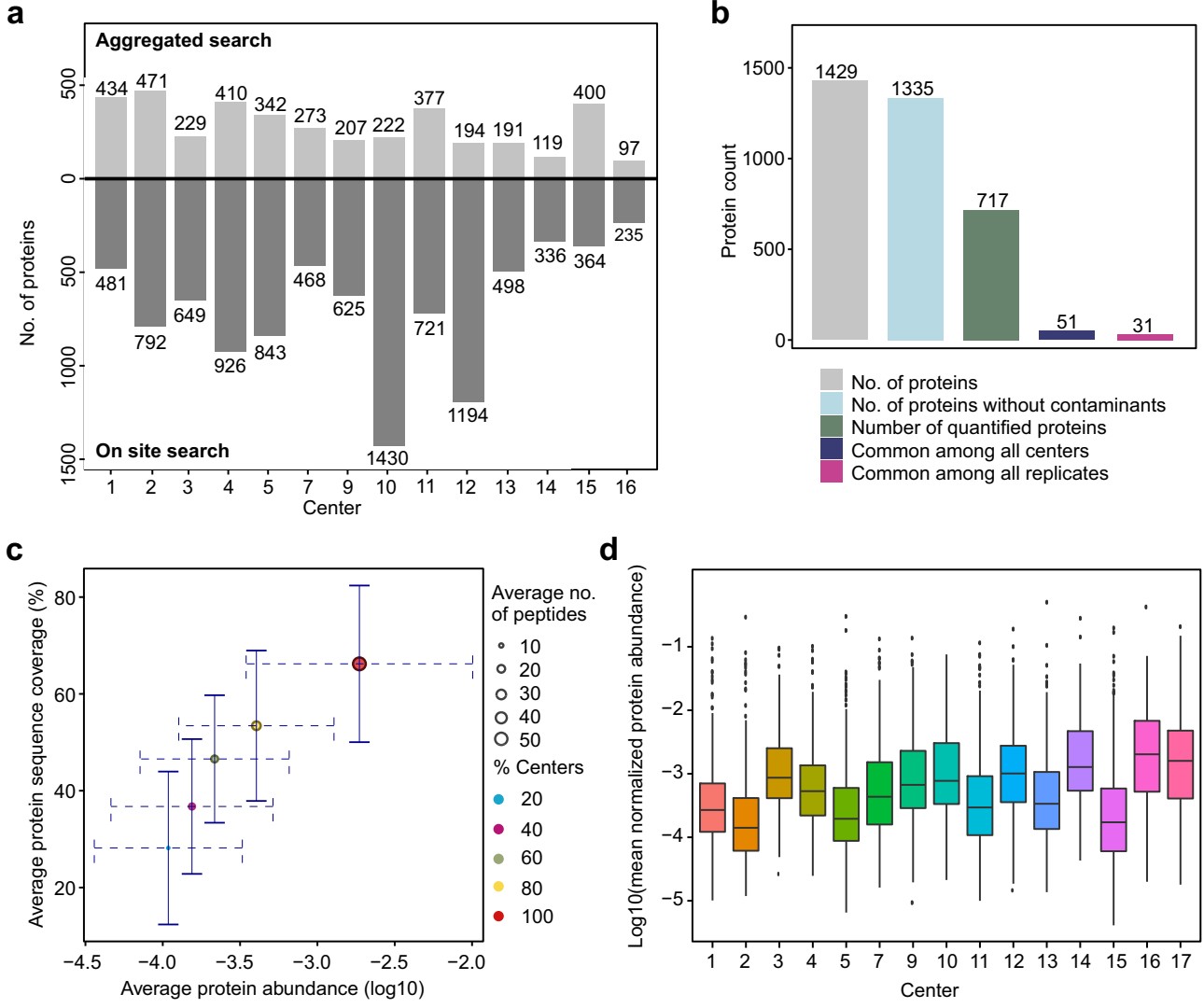

**Fig. 2 | Data overview. a** The number of proteins detected through individual database search by each center vs. those quantified in the uniform database search performed in the current study (light gray for aggregated database search and dark gray for the on-site database search). **b** The number of identified, quantified, and shared proteins across different core facilities. **c** The determining role of protein abundance on protein sequence coverage, the number of detected peptides, and the probability of protein quantification by given centers. Solid error bars on the *y*-axis show the standard deviation of the sequence coverage for each group of centers. Dashed error bars on the *x*-axis show the standard deviation of protein abundances for each group of centers. **d** Distribution of protein-level intensities for the 15 cores (center line, median; box limits contain 50%; upper and lower quartiles, 75 and 25%; maximum, greatest value excluding outliers; minimum, least value excluding outliers; outliers, more than 1.5 times of upper and lower quartiles). All analyses were based on averaging three technical replicates.

included Cys carbamidomethylation as a variable modification and used carbamidomethylated peptides in the quantification of proteins as well. While some centers had included other modifications such as deamidation and phosphorylation in their individual database searches, we included only the routine methionine oxidation and acetylation of protein N-termini as variable modifications, so that the results would be comparable between the centers. A 1% FDR was applied both at the protein and peptide levels, similar to previous studies[19]. As shown in Supplementary Table 1, different centers used 1-10% FDR at the protein and/or peptide level, while some centers did not state the FDR information. While previously some centers had used a semi-specific search, here we only searched for specific tryptic peptides. We also only allowed up to two missed cleavages, which is rather standard and well-accepted in the community. This is while previously several centers had allowed up to 3–5 missed cleavages in their individual search. It is noteworthy that we are only highlighting these variations in parameters as a source of heterogeneity and are applying only parameters that are well-accepted in the community (e.g., no more than 1%

FDR)[20]. This does not undermine the validity of the previous database searches performed individually by different core facilities. The overall workflow is shown in Fig. 1.

## A uniform database search dramatically enhances data homogeneity across distinct centers
The uniform database search identified 1,335 proteins after removing contaminants (Supplementary Data 1). This is while the compilation of individually searched datasets from different core facilities had led to the identification of 4022 proteins, cumulatively[4]. We believe that the significantly higher number of identified proteins in the previous study was partially due to a lack of applying stringent FDR at the protein and peptide levels, as well as using different search engines, sequence databases, and other search parameters such as variable post-translational modifications, enzyme specificity and the number of allowed missed cleavages.

The number of proteins quantified by each core facility is shown in Fig. 2a and compared with the detected proteins in the previous

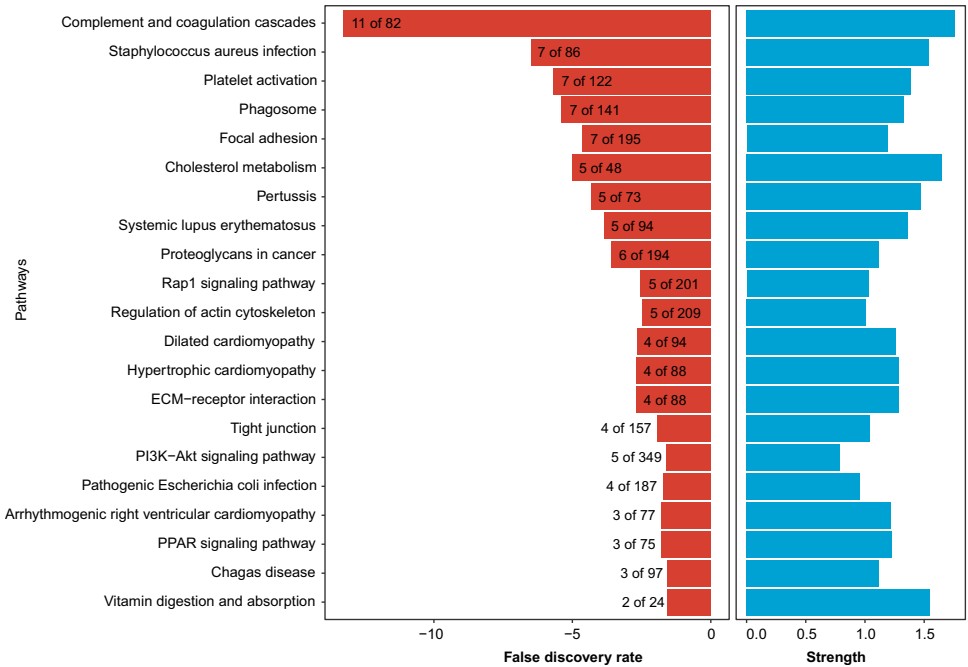

**Fig. 3 | The enriched KEGG pathways for the 51 proteins quantified by all 15 core facilities.** The number of proteins enriched is also given.

study. As expected, the number of proteins for all the centers decreased as a result of the uniform search. Applying a less stringent FDR, semi-specific database search, inclusion of peptides with more than two missed cleavages, and inclusion of non-routine modifications such as phosphorylation are expected to yield a higher number of proteins, in comparison with our uniform search. As a result, compared to the number of detected proteins in individual database searches, in the uniform search, there were more dramatic reductions in the number of quantified proteins for centers that performed the database search using the four above parameters.

Overall, 717 proteins were quantified, 51 (7.1%) of which were shared among all centers (Supplementary Data 2), and 31 (4.3%) were found in all replicates (proteins with NA_count_all_replicates of zero in Supplementary Data 2) (Fig. 2b). In the previous study, only 1.8% of the proteome was shared among 12 core facilities[4]. As expected, the majority of the proteins shared among all centers had a relatively higher abundance, a higher number of detected peptides, and a higher sequence coverage (Fig. 2c; e.g., red and yellow circles show the proteins quantified by 100% and 80% of core facilities). The distribution of the protein-level intensities for the 15 cores is shown in Fig. 2d. We also mapped the 51 shared proteins across all core facilities to the KEGG pathways and the enriched pathways are illustrated in Fig. 3.

A hierarchical clustering of the 717 proteins quantified cumulatively by 15 core facilities in the nanoparticle protein corona is shown in Fig. 4a. The PCA analysis in Fig. 4b also shows that the principal component 1 separates the data by the number of quantified proteins by each core facility. Overall, the cores are mainly separated based on the number of quantified proteins into 3 clusters A, B, and C. We then calculated the number of shared proteins for each cluster. While cluster A core facilities shared 253 proteins, clusters B and C each shared 122 and 57 proteins, respectively (Fig. 4c). Therefore, while 35.3% of proteins were shared among the 5 core facilities in cluster A, the corresponding number for the 6 core facilities in cluster B was 17% and for the 4 core facilities in cluster C was 7.9%. To perform a less biased comparison with the previous study[4], we calculated the shared proteins for centers in clusters A and B (11 centers in total). Cluster A and B shared 116 proteins (16.2% of all proteins). This is a 9-fold improvement compared to the 1.8% shared proteins among 12 centers in our original study[4].

The distribution of protein-level intensities for the shared proteins among the core facilities in each cluster is shown in Fig. 4d, demonstrating that the core facilities in cluster C have mainly quantified the most abundant proteins in the samples (the average protein intensities for the shared proteins in cluster C is higher than B and in turn, that of B is higher than cluster A). These results show that certain core facilities outperform others in quantifying low-abundant proteins and reaching a higher proteome coverage. In comparison with our previous study, these findings indicate that a uniform database search dramatically enhances the number of shared proteins among different core facilities and makes it possible to compare the performance of each center in an unbiased manner. Hypothetically, the shared proteins would also increase if other uniform search parameters were applied.

To further examine the other factors that might be responsible for the observed clustering in Fig. 4a, we investigated the number of times a parameter was used by the cores in each cluster. As shown in Fig. 4e, there was a trend in clustering with respect to the MS system used, digestion mode, and the inclusion of reduction and alkylation steps. Overall, high-end (more recently launched) mass spectrometers excelled in providing the highest proteome coverage, while there were some exceptions. Expectedly, due to lower sample loss, on-bead digestion (though not in all cases) overall provided a higher proteome coverage than in-solution or in-gel digestion, and this might explain why more core facilities opted for on-bead digestion based on their experience. We also investigated the impact of other parameters including gradient duration, variable modifications, LC system, and FDR on the clustering in Fig. 4a. However, none of these parameters had a visible impact on the clustering.

Cys residues constitute 2.3% of the amino acids in the proteome and thus, ≈20% of all tryptic peptides contain at least one Cys[21]. In the absence of reduction and alkylation, such peptides crosslink during or after digestion and are mostly either eliminated during the cleaning process and/or are not found through a routine database search. As such, these peptides cannot be identified/quantified by routine LC-MS/MS analysis. Since 6 core facilities had not indicated Cys reduction and alkylation in the provided protocol (centers 1, 2, 4, 5, 13, and 17), we also calculated the number of quantified Cys-containing peptides for

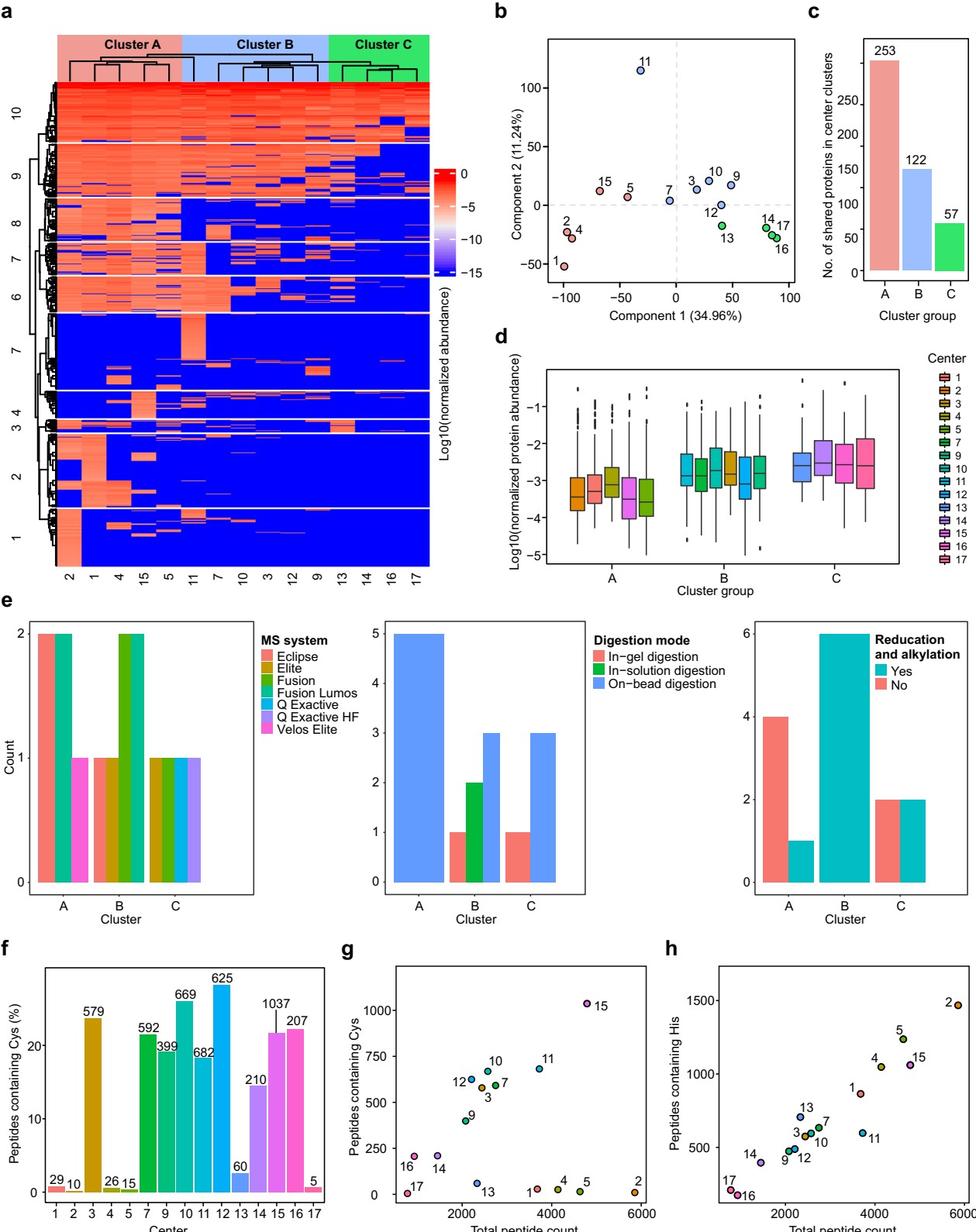

**Fig. 4 | A uniform database search enables comparison of nanoparticle protein corona proteomics data across 15 core facilities. a** Hierarchical clustering of normalized intensities of 717 proteins quantified in the identical nanoparticle protein coronas across 15 core facilities. **b** PCA analysis of the proteomes quantified by 15 core facilities. **c** The number of shared proteins in different clusters identified in hierarchical clustering and PCA analysis of the dataset. The red, blue and green colors in panels **a**–**c** refer to the clusters of centers. **d** The distribution of protein intensities in the different clusters. Boxplot: center line, median; box limits contain 50%; upper and lower quartiles, 75 and 25%; maximum, greatest value excluding outliers; minimum, least value excluding outliers; outliers, more than 1.5 times of upper and lower quartiles. **e** The effect of different parameters on the clustering observed in panel **a**. **f** The percentage of peptides with at least one Cys residue in data obtained from each core facility (numbers on the top of each bar show the actual number of Cys-containing peptides). **g** The number of peptides containing Cys vs. total peptide count. **h** The number of peptides containing His residues vs. total peptide count. All analyses were based on averaging three technical replicates.

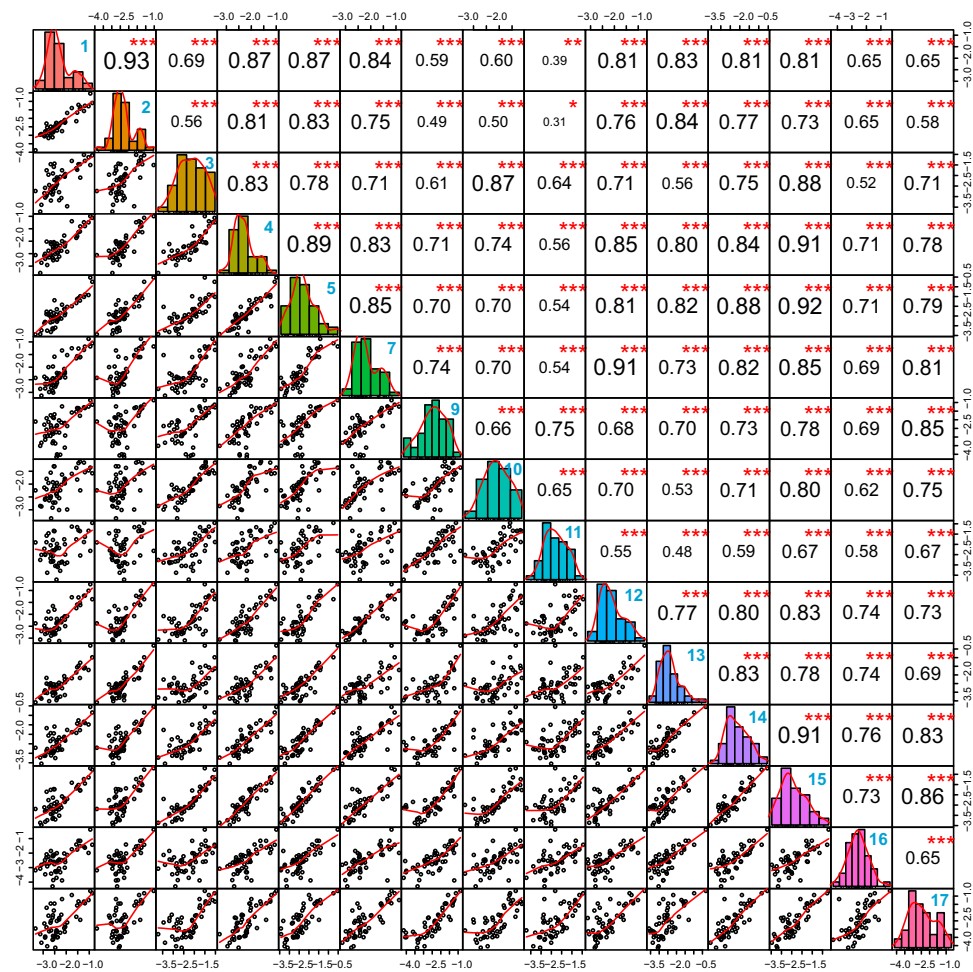

**Fig. 5 | Pearson correlations of normalized protein intensities across 15 cores based on 51 shared proteins.** The mean normalized intensities of three technical replicates for each core were used for correlation analysis. The correlation coefficient is given in black text and core codes are given in bold blue text in the top right corner of the boxes. *** denotes a significance below *the p*-value of 0.001 (two-sided) and is derived from Pearson correlation analysis.

different centers. As shown in Fig. 4f, centers that have included a reduction and alkylation step in their protocols quantified around the expected ~20% Cys-containing peptides. On the other hand, the above 6 centers failed to quantify such peptides. A good correlation is expected between the number of Cys-containing peptides and total peptide count. Such correlation is observed for the centers that performed reduction and alkylation of Cys residues, but not for the 6 centers that skipped this step (Fig. 4g). As a control, we have plotted the number of histidine (His)-containing peptides vs. total peptide count, demonstrating the near-perfect correlation (Fig. 4h). We chose His, as the frequency of His in the proteome is very close to Cys residues[21]. Interestingly, 4 centers that skipped the reduction and alkylation steps, belong to cluster A in Fig. 4a, which quantified the highest number of proteins. However, even in these cases, data quality (peptide number per protein and sequence coverage), as well as the proteome coverage could be further enhanced by the inclusion of Cys-containing peptides. Collectively, this finding shows the necessity of including Cys reduction and alkylation steps in the LC-MS/MS workflow in the prevention of data loss.

### Global similarity of proteomics data across different centers
We then calculated the correlations among the 51 proteins shared by the 15 centers (Fig. 5). This unbiased analysis shows that for the consistently quantified proteins, all cores reported highly comparable data. The data from most core facilities generally show a correlation coefficient higher than 0.7 with those of other facilities. Only

data from core 11 generally show correlations below 0.7 with those of other facilities, that we could not attribute to a single parameter such as digestion time, instrument type, database search, etc. The acceptable correlation levels indicate that the data from each core can be validated within the given proteome coverage and that the variability in data mainly originates from the varying proteome coverage. While the variability can also arise from the LC-MS/MS protocols and workflows, here we show that performing a uniform and standard database search and stringent control of FDR at the protein and peptide levels can dramatically harmonize data across different core facilities.

## Discussion
The results of this study demonstrate that using a uniform and more standard database search can help to homogenize the results of protein corona analysis across different core facilities. The reduction of detected proteins from 4,022 to 1,335 (quantified) proteins across the cores shows that database search parameter settings have a dramatic effect on the search results. The uniform data processing and analysis increases the number of shared proteins among different centers and increases data reproducibility by unifying the sequence database, search parameters, and data cleaning. We also show that while the abundances of shared proteins generally correlate well among different centers, the predominant source of variability originates from various degrees of proteome coverage. In this case, the proteome coverage largely depends on the LC-MS/MS protocol, workflow, the LC

and MS instrument, data extraction, and processing. We further demonstrate that reduction and alkylation are essential steps in sample preparation, and skipping this step can lead to 20% data loss at the peptide level. The lack of detection of Cys-containing peptides can compromise the quality of protein corona analysis in terms of protein sequence coverage and peptide per protein. We further confirm that centers that provide a lower proteome coverage have mainly quantified the more abundant proteins in the corona layers. Proteome coverage is of substantial importance in protein corona research, as the presence and absence of every protein may ultimately influence biomarker selection.

The analyses show that multiple variables in LC-MS/MS protocol, workflow, the LC and MS instrument, data extraction, and processing can account for the outperformance of given centers over others. The MS instrument, digestion mode, and inclusion of reduction and alkylation steps were among the variables that had an effect on the proteome coverage in the current study. However, at times, even several centers skipping Cys reduction and alkylation steps outperformed others who had carried out those steps. The proteome coverage seems to be largely dependent on the workflow, the scientist, and overall, the center performing the experiment. This underscores the importance of good practice in LC-MS/MS sample preparation and underlines the importance of users' expertize as another influential parameter, as stressed before[22].

There are several studies focusing on the technical and instrumental aspects that can introduce heterogeneity in the data retrieved from different centers, including other high-throughput technologies such as RNA sequencing and microarrays[22–29] (though not on nanoparticle protein corona). Other studies have investigated the effects of preanalytical sample processing and storage on plasma proteomics[30–37]. We have already discussed several studies in our previous report[4].

There are certain parameters in database search that can moderately or significantly impact the data output. We believe that the choice of the search engine (software), the sequence database used, the inclusion of several variable modifications, and assigning a specific or semi-specific search can moderately affect the data output, while the FDR rate would probably have the largest impact. While different search engines might yield slightly different outputs, once they have been tested, they are routinely updated, and their outputs can be trusted. The inclusion of additional modifications or performing a semi-specific search would increase the number of quantified peptides (and as such, the number of proteins), but at the expense of the higher rate of false positive discoveries. The selection of a higher FDR cutoff would generally lead to the identification of a larger number of proteins, but in parallel, the number of false positive hits also increases. A higher FDR at the peptide level would allow for a large number of falsely identified peptides, which could later compromise the protein-level inference which is based on peptide-level identifications[38]. A high FDR is especially detrimental for low-abundant proteins that are detected by a few peptides. Therefore, here we will mainly discuss the importance of FDR control.

Some studies have investigated the impact of FDR control in proteomics analysis across different centers. For example, Collins et al.[5] performed a comparative reproducibility analysis of Sequential Window Acquisition of all theoretical fragment ion spectra (SWATH) MS data acquisition among 11 sites worldwide. A set of standard peptides with serial dilutions were spiked into HEK293 cell lysate. While SCIEX TripleTOF 5600/5600+ mass spectrometers were used on all sites, the nano-LC systems were of various models but from the same vendor SCIEX. The study detected a core set of 4,077 proteins that were consistently detected in >80% of the samples. Similar to the outcomes in our study, when the data analysis and FDR control were carried out independently on a site-by-site basis, a reduction of consistently detected proteins was noted among the sites. However, it

should be noted that no sample preparation was performed and thus the variations in the workflows remained minimal. The above example shows that by centralizing sample preparation and data analysis, as well as minimizing variations in the instrumental parameters, it is possible to achieve reproducibility for the majority of proteins in a complex sample.

In comparison with the above study, here the identical protein corona samples were shipped to different centers, where subsequent sample preparation and LC-MS/MS analysis were performed at the discretion of the center and no standardized instrumentation parameters or protocols were requested/provided. Therefore, the only aggregated steps in the current study are the preparation of the protein sample, data extraction, processing, and analysis. Furthermore, the centers in our study had to perform data-dependent acquisition (DDA). Unlike SWATH, DDA acquisition involves stochastic fragment ion (MS2) sampling[39] and the repeatability of peptide sampling is therefore lower[39,40]; i.e., when the number of precursor ions exceeds that of precursor selection cycle, precursor selection becomes stochastic[41].

The Human Proteome Organization (HuPO) test sample working group distributed an equimolar mixture of 20 highly purified recombinant proteins to 27 different labs, which in turn analyzed the samples according to their own routines and protocols without any constraints[40]. Of the 27 labs, only 7 labs correctly reported all 20 proteins. A centralized analysis of the raw data showed that all 20 proteins had been detected in all 27 labs. Missed identifications or false negatives, environmental contamination, problems in database matching, and curation of protein identifications were found as sources of heterogeneity. The study showed that the main variabilities observed in peptide identification and protein assignment were caused by differences in data processing and analysis, rather than data collection.

Here, we demonstrate that by applying a stringent FDR cutoff at the protein and peptide level and unifying the other search parameters such as enzyme specificity, variable post-translational modifications, and consideration of peptides with missed cleavages, the percentage of consistently quantified proteins is increased by a factor of 9, compared to when individual database searches are performed on each site. In comparison with our previous study where an independent database search was performed by different core facilities leading to the identification of 4022 proteins, here we identified 1429 proteins and quantified only 717 proteins.

We believe that the challenging nature of plasma proteomics in general, is one of the main reasons for lower rates of consistently detected proteins in the corona layers, compared to other types of proteomics analysis (e.g., cells and tissues). Similarly, in a previous investigation, the authors reanalyzed data from 178 experiments from 2005–2017, showing that only 50% of the studies reported the 500 most abundant plasma proteins[12].

In summary, we revealed that the use of a uniform database search provides an opportunity for taking measures in best practices and quality control in protein corona research using LC-MS/MS. This approach paves the way to harmonize data analysis of protein corona outcomes, enabling stakeholders to perform meta-analyses of proteomics data in the existing literature. It seeks to minimize conflicts and discrepancies that have arisen due to differences in sample preparation and workflow across labs[3]. Enhancing reproducibility and proteome coverage in protein corona research can accelerate the successful clinical translations of nanomedicine technologies both in diagnosis and therapeutic applications.

## Methods

### LC-MS/MS sample preparation

Details of sample preparation and LC-MS/MS analysis can be found in the Supplementary Information.

## LC-MS/MS data processing and analysis

The raw LC-MS/MS data were analyzed by MaxQuant[42], version 2.2.0.0. The Andromeda search engine[43] matched MS/MS data against the UniProt complete proteome database (human, 20,401 entries without isoforms, downloaded on December 11th, 2022). Since some centers had not specified if alkylation of Cys residues had been performed, we included Cys carbamidomethylation as a variable modification (and used for protein quantification), along with methionine oxidation and acetylation of protein N-termini. Trypsin/P was selected as enzyme specificity. No more than two missed cleavages were allowed. A 1% FDR was used as a filter at both protein and peptide levels. The first search tolerance was 20 ppm (default) and main search tolerance was 4.5 ppm (default), and the minimum peptide length was 7 residues. Due to the different lengths of LC gradients across different core facilities, the match-between-run option was not activated. For all other parameters, the default MaxQuant settings were used.

## Data analysis

First, for each core, data were normalized by total protein intensity in each technical replicate. Data analysis was performed using R project version 4.1.0.

## Statistics and reproducibility

All centers performed a triplicate analysis of a given aliquot.

## Reporting summary

Further information on research design is available in the Nature Portfolio Reporting Summary linked to this article.

## Data availability

The data are from our previous study[4]. Due to the blinding of core names in the current study, and since the MS .raw files can be traced, the .raw data and associated individual data files are available upon request from corresponding authors (A.A.S. and M.M.). The processed datasets are provided as Supplementary Data files.

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

## Acknowledgements

This work is supported by the U.S. National Institute of Diabetes and Digestive and Kidney Diseases award DK131417 (M.M.). Mass spectrometry data were collected in the University of Cincinnati Proteomics Laboratory under the direction of K.D. Greis, Ph.D. on a system that was obtained in part through an NIH high-end instrumentation grant (S10OD026717-01). A.A.S. was supported by the Swedish Research Council (grant 2020-00687) and the Swiss National Science Foundation (Ambizione Fellowship, SNSF grant number: PZ00P3_216203). We acknowledge all 17 core facilities for their efforts and contributions to the current study. We acknowledge the support of a Burroughs Wellcome Fund Career Award at the Scientific Interface (CASI) (M.P.L.), a Dreyfus Foundation award (M.P.L.), the Philomathia Foundation (M.P.L.), an NIH MIRA award R35GM128922 (M.P.L.), an NIH R21 NIDA award 1R03DA052810 (M.P.L.), an NSF CAREER award 2046159 (M.P.L.), an NSF CBET award 1733575 (to M.P.L.), a CZI imaging award (M.P.L.), a Sloan Foundation Award (M.P.L.), a McKnight Foundation award (M.P.L.), a Simons Foundation Award (M.P.L.), a Moore Foundation Award (M.P.L.), and a Schmidt Foundation Award (M.P.L.). M.P.L. is a Chan Zuckerberg Biohub investigator, a Hellen Wills Neuroscience Institute Investigator, and an IGI Investigator.

## Author contributions

Conceptualization A.A.S. and M.M.; project organization, resources, and funding acquisition, A.A.S. and M.M.; experimental data extraction and processing, A.A.A.; data analysis and visualization, H.G., M.J., and A.A.S.; writing—original draft, A.A.S., H.G., A.A.A., M.J., E.V., M.P.L., and M.M.; writing, review & editing, all co-authors.

## Competing interests

M.M. discloses that (1) he is a co-founder and director of the Academic Parity Movement (www.paritymovement.org), a non-profit organization dedicated to addressing academic discrimination, violence, and incivility; (2) he is a co-founder of and shareholder in Targets' Tip Corp.; and (3) he receives royalties/honoraria for his published books, plenary lectures, and licensed patents. The remaining authors declare no competing interests.
