## [Peer Review File · Nature Communications]

Reviewers' Comments:

Reviewer #1:

Remarks to the Author:

Hassan Gharibi and colleagues, describe the harmonization of proteomics data generated by mass spectrometry analysis of nanoparticles protein corona in a number of core facilities across USA. In respect to the previous publication (ref 4), the team seems moving into a path of SOPs harmonization. Here the harmonization regards only the workflow for data analysis, but this study could provide new insight into the composition of protein corona and keeping paving the way for future robust analysis of protein corona.

However, before publication, the authors should revise some aspect of the paper, in particular the discussion of the data.

1. The current title is "A uniform database search dramatically homogenizes nanoparticle protein corona data across proteomics core facilities". I suggest to substitute the word "homogenize" with harmonization. For example : " A common data processing pipeline enables harmonized nanoparticle protein corona analysis across proteomics core facilities"

A few suggestions for the discussion:

2. The authors here confirm that proteomics data search and analysis is extremely important to achieve consistent protein coverage as shown also in previous studies (ref. 40). Nevertheless, the presence of three independent clusters shows that there are obviously other variables, behind data analysis, that will need to be harmonized. The authors should this aspect more deeply. Are there any common aspects in the analysis workflow for the facilities in each cluster?

3. It is well known that when performing immunocapture based on beads, a big part of the proteins identified in the samples are actually contaminants or background proteins nonspecifically purified/identified. In respect to that, it could be hypothesized that, some of the facilities identifying the largest number of proteins, which are not reproducible elsewhere are actually identifying simply more "background" After data analysis harmonization, 51 proteins were consistently identified across all the facilities. Could this result be considered already satisfactory? How many proteins are expected to be found to be specifically part of protein corona? Will this protein change for different types of nanoparticles?

4. The 51 proteins should be reported and discussed. Is there anything biologically relevant and/or of interest among the 51 proteins consistently identified?

5. Minor comment. Figure 2a is not immediate to understand. Better to use the same color for all the core facilities, and two colors only to indicate previous versus new protein identifications

Reviewer #2:

Remarks to the Author:

In this study, Gharibi et al. demonstrated that a uniform database search improved protein data searches across 15 proteomic facilities.

Major issues:

1) The novelty is lacking. The conclusions are obvious and not original. Indeed, the standardized proteomic pipelines generate more consistent proteomic data across different facilities.

2) The manuscript describes a re-analysis of the previously published datasets, with no novel data.

3) In fact, some facilities avoided Cys reduction and alkylation, so the reported differences in data would be expected.

4) "Collectively, this finding shows the necessity of including Cys reduction and alkylation steps in the LC-MS/MS workflow in prevention of data loss."

"We further demonstrate 236 that reduction and alkylation is an essential step in sample preparation, and skipping this step can 237 theoretically lead to 20% data loss at the peptide level."

These two conclusions are obvious. Proper sample preparation is essential for high-quality proteomic analysis.

5) "Protein corona" is an ambiguous term for the non-specific interactions. The manuscript does not provide any details and states only the ambiguous term "nanoparticles".

Reviewer #3:

Remarks to the Author:

In the manuscript by Hassan Gharibi and Morteza Mahmoudi entitled "A uniform database search dramatically homogenizes nanoparticle protein corona data across proteomics core facilities" the authors describe how they used consisting data from a previous paper and improved data quality assessment by re-analyzing the primary data by a consistent workflow and a stringent set of parameters for the search. Especially setting the FDR to 1%, using the same database and pointing out that the amount of proteome coverage influences the results is important for the community. Also the description that reduction and alkylation steps in the protocols results in different outcomes is important for the reader.

I have just one minor remark for this paper:

1. The authors should point to the reduction and alkylation step as being important also in the abstract.

Otherwise the paper is very well written, concise and help to read her in the community to improve consistency.

Response to reviewer comments

Reviewer #1 (Remarks to the Author):

Hassan Gharibi and colleagues, describe the harmonization of proteomics data generated by mass spectrometry analysis of nanoparticles protein corona in a number of core facilities across USA.

In respect to the previous publication (ref 4), the team seems moving into a path of SOPs harmonization. Here the harmonization regards only the workflow for data analysis, but this study could provide new insight into the composition of protein corona and keeping paving the way for future robust analysis of protein corona.

However, before publication, the authors should revise some aspect of the paper, in particular the discussion of the data.

Response: We thank the reviewer for their positive assessment of our work.

1. The current title is “A uniform database search dramatically homogenizes nanoparticle protein corona data across proteomics core facilities”. I suggest to substitute the word “homogenize” with harmonization. For example :” A common data processing pipeline enables harmonized nanoparticle protein corona analysis across proteomics core facilities”

Response: We thank the reviewer for the wise comment; we have now updated the title in the revised manuscript as follows “A uniform data processing pipeline enables harmonized nanoparticle protein corona analysis across proteomics core facilities”.

A few suggestions for the discussion:

2. The authors here confirm that proteomics data search and analysis is extremely important to achieve consistent protein coverage as shown also in previous studies (ref. 40). Nevertheless, the presence of three independent clusters shows that there are obviously other variables, behind data analysis, that will need to be harmonized. The authors should discuss this aspect more deeply. Are there any common aspects in the analysis workflow for the facilities in each cluster?

Response: We thank the reviewer for this constructive comment. We analyzed the impact of all parameters on the clustering and the results are summarized in the new **Fig. 3e**. No trends were observed for the other parameters described in **Supplementary Table 1**. A discussion is also added around this finding.

3. It is well known that when performing immunocapture based on beads, a big part of the proteins identified in the samples are actually contaminants or background proteins nonspecifically purified/identified. In respect to that, it could be hypothesized that, some of the facilities identifying the largest number of proteins, which are not reproducible elsewhere are actually identifying simply more “background” After data analysis harmonization, 51 proteins were consistently identified across all the facilities. Could this result be considered already satisfactory? How many proteins are expected to be found to be specifically part of protein corona? Will this protein change for different types of nanoparticles?

Response: All contaminants were removed prior to the analysis. While we cannot rule out that some of these proteins might be background, since the samples were identical, in the best case, all proteins should have been quantified by all cores.

4. The 51 proteins should be reported and discussed. Is there anything biologically relevant and/or of interest among the 51 proteins consistently identified?

Response: We have now included two supplementary data tables with all 1335 proteins, 51 shared proteins across all centers and 31 shared proteins across all replicates (along with normalized intensities). We have also added **Fig 3** showing the pathways that are enriched for the 51 shared proteins.

5. Minor comment. Figure 2a is not immediate to understand. Better to use the same color for all the core facilities, and two colors only to indicate previous versus new protein identifications.

Response: The figure is now modified according to the reviewer's comment.

Reviewer #2 (Remarks to the Author):

In this study, Gharibi et al. demonstrated that a uniform database search improved protein data searches across 15 proteomic facilities.

Major issues:

1) The novelty is lacking. The conclusions are obvious and not original. Indeed, the standardized proteomic pipelines generate more consistent proteomic data across different facilities.

Response: In this paper, we aimed to show that the lack of standardization in database search parameters can lead to inconsistency in the data outcomes obtained by different cores. While this should be common sense, we show that non-optimal data processing can lead to false discoveries and data heterogeneity.

2) The manuscript describes a re-analysis of the previously published datasets, with no novel data.

Response: Centralized reanalysis of the data in the previous study was indicated by the reviewers of the previous paper. This was not a trivial task. Therefore, we decided to have a focused study on this issue.

3) In fact, some facilities avoided Cys reduction and alkylation, so the reported differences in data would be expected.

Response: Yes, again this should be common sense and reduction and alkylation steps should be included. But seemingly, some cores opt not to include these steps, leading to quantification of fewer peptides. We hope that our findings will prompt other researchers to follow the common guidelines.

4) "Collectively, this finding shows the necessity of including Cys reduction and alkylation steps in the LC-MS/MS workflow in prevention of data loss." "We further demonstrate that reduction and alkylation is an essential step in sample preparation, and skipping this step can theoretically lead to 20% data loss at the peptide level."

These two conclusions are obvious. Proper sample preparation is essential for high-quality proteomic analysis.

Response: We agree with the reviewer. We were also surprised to find out that some cores did not include reduction and alkylation in their protocols. We noted this issue during data analysis for this paper and decided to investigate this further. The reviewer is correct, and it should be common sense to include reduction and alkylation, but apparently it is not.

5) “Protein corona” is an ambiguous term for the non-specific interactions. The manuscript does not provide any details and states only the ambiguous term “nanoparticles”.

Response: Full details on the protein corona formation at the surface of well-characterized polystyrene nanoparticles have been conducted and reported in the original report. The “protein corona” and the nature of its interactions with nanoparticles are well-recognized/studied term in nanomedicine [please see our recent review for details: Mahmoudi, M., Landry, M.P., Moore, A. and R. Coreas The protein corona from nanomedicine to environmental science. *Nat Rev Mater* 8, 422–438 (2023). <https://doi.org/10.1038/s41578-023-00552-2>].

Reviewer #3 (Remarks to the Author):

In the manuscript by Hassan Gharibi and Morteza Mahmoudi entitled "A uniform database search dramatically homogenizes nanoparticle protein corona data across proteomics core facilities" the authors describe how they used consisting data from a previous paper and improved data quality assessment by re-analyzing the primary data by a consistent workflow and a stringent set of parameters for the search. Especially setting the FDR to 1%, using the same database and pointing out that the amount of proteome coverage influences the results is important for the community. Also the description that reduction and alkylation steps in the protocols results in different outcomes is important for the reader.

Response: We thank the reviewer for their positive assessment.

I have just one minor remark for this paper:

1. The authors should point to the reduction and alkylation step as being important also in the abstract.

Response: This statement was added to the abstract: “Furthermore, we note that reduction and alkylation are important steps in sample processing and as expected, omitting these steps reduces the number of total quantified peptides by 20%.”

Otherwise the paper is very well written, concise and help to read her in the community to improve consistency.

Reviewers' Comments:

Reviewer #1:

Remarks to the Author:

The authors answered to all my points including also new figures.

The only point that remains is the "background" issue.

Indeed even if the samples has been prepared identically, the simple fact that different facilities applied different digestion procedure and instruments may cause the identification of a variable numbers of proteins. Indeed some instruments may be more sensitive than others, and detect more proteins.

This paper highlights the importance of consistency in sample preparation and data analysis, and I found it interesting and useful to keep raising the awareness on need for reproducibility, so it is worth publishing.

However, I would suggest to perform a study, using the same protocol for digestion and same system for sample analysis and then data analysis.

Response to reviewer comments

Reviewer #1 (Remarks to the Author):

The authors answered to all my points including also new figures.

The only point that remains is the "background" issue.

Indeed even if the samples has been prepared identically, the simple fact that different facilities applied different digestion procedure and instruments may cause the identification of a variable numbers of proteins. Indeed some instruments may be more sensitive than others, and detect more proteins.

This paper highlights the importance of consistency in sample preparation and data analysis, and I found it interesting and useful to keep raising the awareness on need for reproducibility, so it is worth publishing.

However, I would suggest to perform a study, using the same protocol for digestion and same system for sample analysis and then data analysis.

Response: We would like to extend our sincere gratitude for the valuable comments provided by the reviewer. We are delighted to see that the reviewer found our responses and revisions satisfactory.

As we also show in Fig. 4g, there are no strong dependencies between the type of mass spectrometer used and the number of proteins that are quantified. This is exemplified by the oldest instruments, Orbitrap Velos Elite and Orbitrap Elite that appear in clusters A and B, while the more recent Orbitrap Fusion or Q Exactive HF appear in cluster C. Sensitivity is only one of the issues that is impacting the data outcome and other parameters such as sample preparation should also be systematically investigated.

We still acknowledge the importance of the suggestion of new set of experiments using the same protocol for digestion and the same system for sample analysis; however, adding such results may not align with the current aims and objectives of our paper. In a recent publication (Nature Communications, <https://www.nature.com/articles/s41467-022-34438-8>), we highlighted a significant variation in protein corona outcomes for identical nanoparticles across different core facilities. This diversity in protein corona datasets has made it challenging to compare results across independent studies. To address this issue, we proposed two distinct methods:

Unified Database Search, Data Processing and Analysis (which is the focus of the current submitted paper): This approach aims to harmonize data analysis of protein corona outcomes, enabling stakeholders to perform meta-analyses of proteomics data in the existing literature. It seeks to resolve conflicts and discrepancies that have arisen due to differences in sample preparation and workflow across labs (for more details, please refer to <https://onlinelibrary.wiley.com/doi/full/10.1002/sml.202301838>). We have now added full argument about this aim to the revised manuscript to make it more clear to the readers.

Standard Protocols and Workflow Unification: We have already performed this study (sending the identical batch of the in-house prepared protein corona peptides to 15 different mass centers) and are in the process of data analysis. The main goal here is to develop standard protocols and unifying workflows for peptide preparation and analysis to improve protein corona proteomics reproducibility. This work is intended to be submitted to Nature Communications in the coming months.

It is essential to emphasize that these two strategies serve distinct purposes. The current strategy, focusing on uniform data analysis, aims to enhance the robustness of proteomics data available in the literature and minimize misinterpretations. On the other hand, the upcoming paper, which emphasizes the use of consistent sample preparation and instruments, will underscore the need for establishing gold standard mass spectrometry protocols for protein corona. This is crucial for addressing the reproducibility challenges in future research.

Given the unique goals and objectives of each manuscript, we believe that combining these two strategies may confuse readers and dilute the focus on the primary claims of each work.

Reviewers' Comments:

Reviewer #1:

Remarks to the Author:

I agree with the authors that there is no need to include further experiments in this paper.
I am satisfied with the answer and I do not have further comments.